# Neural Algorithmic Reasoning Without Intermediate Supervision

**Gleb Rodionov**
Yandex Research
Moscow, Russia
rodionovgleb@yandex-team.ru

**Liudmila Prokhorenkova**
Yandex Research
Amsterdam, The Netherlands
ostroumova-la@yandex-team.ru

## Abstract

Neural algorithmic reasoning is an emerging area of machine learning focusing on building models that can imitate the execution of classic algorithms, such as sorting, shortest paths, etc. One of the main challenges is to learn algorithms that are able to generalize to out-of-distribution data, in particular with significantly larger input sizes. Recent work on this problem has demonstrated the advantages of learning algorithms step-by-step, giving models access to all intermediate steps of the original algorithm. In this work, we instead focus on learning neural algorithmic reasoning only from the input-output pairs without appealing to the intermediate supervision. We propose simple but effective architectural improvements and also build a self-supervised objective that can regularise intermediate computations of the model without access to the algorithm trajectory. We demonstrate that our approach is competitive to its trajectory-supervised counterpart on tasks from the CLRS Algorithmic Reasoning Benchmark and achieves new state-of-the-art results for several problems, including sorting, where we obtain significant improvements. Thus, learning without intermediate supervision is a promising direction for further research on neural reasoners.

## 1 Introduction

Building neural networks that can reason has been a desired goal for decades of research. While the ability to reason can be defined and measured in various ways, most of them represent a challenge for current models, solving which can allow the community to use such models for complex real-world problems. Significant progress has been achieved in the reasoning abilities of language models (Touvron et al., 2023, Hoffmann et al., 2022), where reasoning can be described as the ability to answer questions or solve logical and mathematical problems in formal (Polu and Sutskever, 2020) or natural language (Lewkowycz et al., 2022, Jiang et al., 2023). However, the essential part of this progress is inseparable from the large scale of such models, which is needed to abstract over complex language structures (Wei et al., 2022).

Recent research considers classic algorithms as a universal tool to define reasoning on simple instances like sets of objects and relations between them, making algorithmic modeling a popular domain for testing neural networks and highlighting current reasoning limitations of existing architectures (Zaremba and Sutskever, 2014, Kaiser and Sutskever, 2015, Trask et al., 2018, Kool et al., 2019, Dwivedi et al., 2023, Yan et al., 2020, Chen et al., 2020b).

*Neural algorithmic reasoning* is an area of machine learning focusing on building models that can execute classic algorithms (Veličković and Blundell, 2021). The motivation is to combine the advantages of neural networks (processing raw and noisy input data) with the appealing properties of algorithms (theoretical guarantees and strong generalization). Assuming that we have a neural network capable of solving a classic algorithmic task, we can incorporate it into a more complex

37th Conference on Neural Information Processing Systems (NeurIPS 2023).

pipeline and train end-to-end. For instance, if we have a neural solver aligned to the shortest path problem, it can be used as a building block for a routing system that accounts for complex and dynamically changing traffic conditions.

The main challenge of neural algorithmic reasoning is the requirement of out-of-distribution (OOD) generalization. Since algorithms have theoretical guarantees to work correctly on any input from the domain of applicability, the difference between train and test data distributions for neural algorithmic reasoners is potentially unlimited. Usually, the model is required to perform well on much larger input sizes compared to the instances used for training. This feature distinguishes neural algorithmic reasoning from standard deep learning tasks: neural networks are known to perform well on unseen data when it has the same distribution as the train data, but changes in the distribution may significantly drop the performance. This challenge is far from being solved yet; for example, as shown by Mahdavi et al. (2023), even achieving the almost perfect test score, neural reasoners can rely on implicit dependencies between train and test data instead of executing the desired algorithm, which was demonstrated by changing the test data generation procedure.

The area of neural algorithmic reasoning has been developing rapidly in recent years, and graph neural networks (GNNs) have proven their applicability in this area in various contexts. In particular, Veličković et al. (2020) and Xu et al. (2020) analyzed the ability of GNNs to imitate classic graph algorithms and explored useful inductive biases. Further, Dudzik and Veličković (2022) showed that GNNs align well with dynamic programming. Also, several different setups demonstrated that neural reasoners benefit from learning related algorithms simultaneously, explicitly (Georgiev and Lió, 2020, Numeroso et al., 2023) or implicitly (Xhonneux et al., 2021, Ibarz et al., 2022) reusing common subroutines. A significant step forward in developing neural algorithmic reasoners was made with the proposal of the CLRS benchmark (Veličković et al., 2022) that covers 30 diverse algorithmic problems. In this benchmark, graphs are used as a general tool to encode data since they straightforwardly adapt to varying input sizes.

An important ingredient of the CLRS benchmark is the presence of *hints* that represent intermediate steps of the algorithm's execution. With hint-based supervision, the model is forced to imitate the exact trajectory of a given algorithm. Such supervision is considered a fundamental ingredient for progress in neural algorithmic reasoning (Veličković et al., 2022).

In this paper, we argue that there are several disadvantages of using hints. For example, we demonstrate that the model trained without hint supervision has a greater tendency to do parallel processing, thus following the original sequential trajectory can be suboptimal. We also show that while the model trained with supervision on hints demonstrates the alignment to the original algorithm trajectory, it could not capture the exact dependency between the hint sequence and the final result of the execution.

Motivated by that, we investigate whether it is possible to train a competitive model using only input-output pairs. To improve OOD generalization, we propose several modifications to the standard approach. As a first step, we propose a simple architectural improvement of the no-hint regime that makes it better aligned with the hint-based version and, at the same time, significantly boosts the performance compared to the typically used architecture. More importantly, for several tasks, we add a contrastive regularization term that forces similar representations for the inputs having the same execution trajectories. This turns out to be a useful inductive bias for neural algorithmic reasoners.

The experiments show that our approach is competitive with the current state-of-the-art results relying on intermediate supervision. Moreover, for some of the problems, we achieve the best known performance: for instance, we get the F1 score $98.7\%$ for the sorting, which significantly improves over the previously known winner with $95.2\%$.

We hope our work will encourage further investigation of neural algorithmic reasoners without intermediate supervision.

## 2   Background

Our work follows the setup of the recently proposed CLRS Algorithmic Reasoning Benchmark (CLRS) (Veličković et al., 2022). This benchmark represents a set of 30 diverse algorithmic problems. CLRS proposes graphs as a general way to encode data since any particular algorithm can be expressed as manipulations over a set of objects and relations between them. For example, for the Breadth-first search (BFS), the input represents the original graph with a masked starting node, and the output

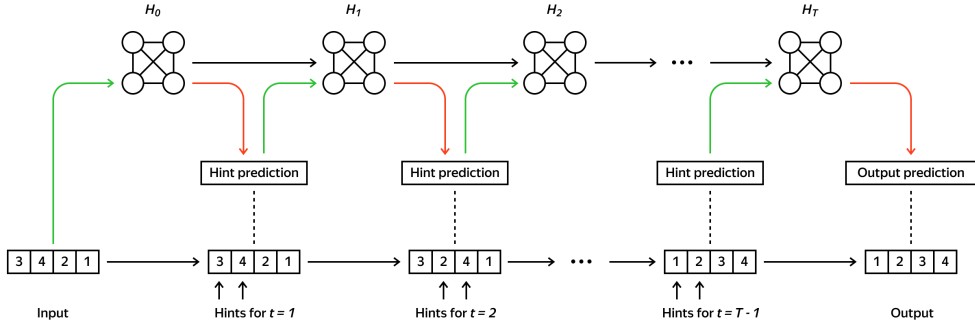

Figure 1: Hints usage diagram for the Bubble sort. Green lines represent the encoders, and red — the decoders. The dotted lines illustrate the supervision. Hints at each step represent the current order of the elements and two indicated elements for the last comparison.

of the algorithm (traversal order of the graph) can be encoded via predicting the predecessor node for any node. For non-graph tasks, CLRS proposes an approach to derive the graph structure. For example, for sorting algorithms, each element of the array can be treated as a separate node, endowed with position scalar input for indexing, and the output is encoded as a pointer from each node to its predecessor in the sorted order.

An important component of the CLRS benchmark is *hints*, which represent the decomposition of the algorithm's execution into intermediate steps. Each hint contains all the necessary information to define the next execution step. For example, for Bubble sort, a hint encodes the current order of the elements and two indexes of the elements for the next comparison. We note that each algorithm defines a unique way of solving the problem, which can be learned separately.

Each transition between the consecutive trajectory states defines a new task, which can be learned separately or simultaneously. Recent work has demonstrated the utility of the encode-process-decode paradigm for learning the whole trajectory of the algorithm at once (Veličković et al., 2022, Ibarz et al., 2022) (Figure 1). At each time step $t$, all input data are encoded with simple encoders to a high-dimensional (we use $d = 128$) latent space, producing vectors $(x_i^t) \in \mathbb{R}^{n \times d}$ representing nodes, $(e_{ij}^t) \in \mathbb{R}^{n \times n \times d}$ edges, and $(g^t) \in \mathbb{R}^d$ graph features, where $n$ is the task size (nodes count).

Then, a processor, usually a graph neural network (Veličković et al., 2022, Ibarz et al., 2022) or a transformer (Diao and Loynd, 2023, Mahdavi et al., 2023), operates in this latent space, executing a transition to the next step. The processor computes the hidden states $h_i^t$ for the nodes (and optionally edges) from the combination of the encoded inputs and hidden states $h_i^{t-1}$ from the previous step, where $h_i^0 = 0$. After this transition, the latent vectors are decoded to predict the hints for the current step. Then, the current step hints prediction can be discarded or fed back into the processor model for the next step, similarly to the encoded input, allowing the model to follow the predefined dynamic of the hints sequence.

At the last step of the trajectory, decoders project the output prediction to the problem. Similarly to the encoders, the decoders rely mainly on a linear layer for each hint and output. The decoders are also endowed with a mechanism for computing the pairwise node similarities when appropriate, which allows one to unify a method for predicting node-to-node pointers for graphs with different sizes. We refer to Veličković et al. (2022) for a more detailed explanation. It was also recently shown that the processor can be shared across tasks and thus reuse algorithmic subroutines (Ibarz et al., 2022).

Another recent work (Bevilacqua et al., 2023) shows a significant improvement in size generalization via a novel way of using hints called Hint-ReLIC. The method comes from the observation that for some algorithms, there exist multiple different inputs on which certain steps of the algorithm will perform identically. Designing causal structure on the algorithm's trajectory, the authors have demonstrated that the execution of each step depends only on a small part of the current snapshot, meaning that the invariance under changes of irrelevant parts can serve as a regularization for the model. For example, the Bubble sort at each step compares only two elements of the array, so any

changes of other elements or adding new elements to the array should not affect the execution of the given step. Such regularization forces better alignment of the model to the relevant computations, improving size generalization. Invariance is forced via a self-supervised objective which is a modified version of the ReLIC algorithm (Mitrovic et al., 2021).

## 3 Neural algorithmic reasoning without hints

**Motivation**   There are several disadvantages of using hints in neural algorithmic reasoning. First, hints need to be carefully designed separately for each task and each particular algorithm. This may require a significant amount of human effort: e.g., the hint at the next step should be predictable by the hint from the previous step using a single round of the message passing (Veličković et al., 2022). Moreover, as this example shows, different model architectures may benefit from differently designed hints: for a processor based on a graph transformer, the optimal hints can be different. Mahdavi et al. (2023) demonstrated the comparison of the performances for different GNN architectures depending on hint usage, showing that for some algorithms the question of using hints is faced by each architecture separately.

As discussed above, hints force the model to follow a particular trajectory. While this may help the interpretability and generalization of the trained model, following the algorithm's steps can be suboptimal for a neural network architecture. For instance, it may force the model to learn sequential steps, while parallel processing can be preferable. Moreover, a model trained with direct supervision on hints could not necessarily capture the dependencies between hints and the output of the algorithm.

To demonstrate this disadvantage with a practical example, we compared the dynamics of the predictions for the model trained on the Insertion sort problem depending on hint usage, see Figure 2. For each model, we evaluated the intermediate predictions after each processor step and compared them with the final predictions of the model. We note that for hint-based models we have two decoders for predictions: the hint decoder and the output decoder, both of which can be used for this purpose.

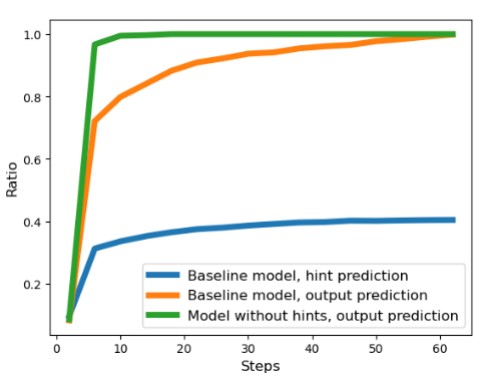

We see that the model trained with supervision on the trajectory for Insertion sort struggles to learn the relation between the intermediate predictions of the pointers and the output predictions (blue line). In other words, the model potentially can consider the hint sequence (and even each transition between hints) as a separate task, not related to the output of the algorithm directly.

Figure 2: The ratio of pointers predicted after a given number of steps which are equal to the final model output. Insertion sort task, test data (64 nodes).

We also see that the execution trajectory for the models with hints inherits some sequential order of intermediate updates (orange line). The model without hints has a much stronger tendency to do parallel processing — almost all output predictions are obtained on the first steps of the execution.

Finally, we note that if a model is not restricted to a particular trajectory, it may potentially learn new ways to solve the problem, and thus by interpreting its computations, we may get new insights into the algorithms' design.

**Simple improvements of the no-hint mode**   Let us describe several changes to the no-hint mode that make the comparison with hint-based methods more straightforward and, at the same time, significantly boost the performance compared to the typically used no-hint regime.

First, we propose an architectural modification of the no-hint mode that makes its computational graph more similar to the standard regime with hints. As described in Section 2, in the default implementation of the encode-process-decode paradigm of the CLRS benchmark, all hint predictions from the current step are fed back to the processor model (as illustrated in Figure 1). Thus, in addition

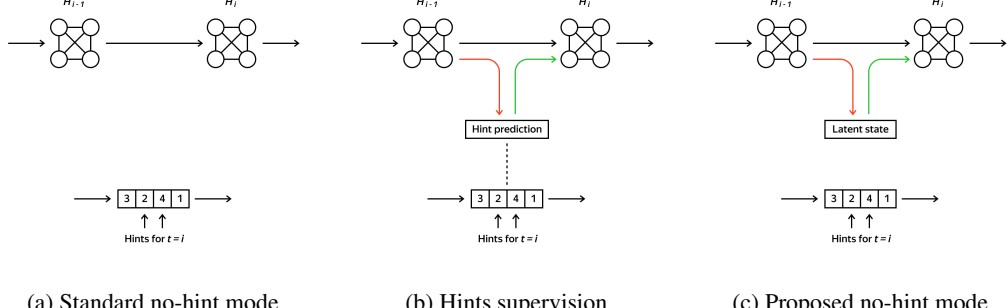

(a) Standard no-hint mode      (b) Hints supervision      (c) Proposed no-hint mode

Figure 3: One step of the algorithm's execution depending on the different hint usage modes for the Bubble sort algorithm.

to the intermediate supervision, hint predictions are also used in the next computational step. So, the transition of the hidden states from step $t-1$ to step $t$ uses not only the encoded inputs and hidden states $h_i^{t-1}$ but also the current hint predictions, which are derived from the hidden state via additional computations.

However, for the no-hint version, as there are no intermediate steps to supervise, hints are not predicted and not fed back to the model. While this approach seems natural, it leads to significant computation graph differences compared to the model with hints. We note that the computation of hint predictions and encoding them to the processor in the encoded-decoded mode can be expressed as an additional message-passing step over the nodes, edges, and graph features (which can differ from the message-passing procedure of the processor, if any). Thus, it can be reasonable to preserve this part even in the no-hint regime. So, we propose to run the no-hint model in the encoded-decoded[1] mode but do not supervise hint predictions, allowing the model to construct useful representations and use them in the next steps. This modification makes the computation graphs similar and also noticeably improves the performance of the no-hint regime. We refer to Figure 3 for the illustration of the proposed changes.

Second, we note that all algorithms (and, therefore, hint sequences) assume sequential data representation. However, for some algorithms, only the intermediate steps depend on the position encoding of nodes, while the outputs are independent of the position scalars. For example, the output for the sorting algorithms is represented as pointers to the predecessor in the sorted order (and the node with the minimal value points to itself). Positions are only needed to tie-break identical elements, but due to the data generation in the CLRS benchmark, such collisions do not occur. To avoid unnecessary dependency of models on the position encodings, we do not use them if the output is independent of positions. For some algorithms, however, the positions are needed: for instance, the Depth-first search traversal depends on vertex indexing. Thus, we keep the position scalars only for the DFS, SCC, and Topological sort problems.

## 4 Self-supervised regularization

### 4.1 Reducing the search space

In this section, we propose a self-supervised objective that can regularise intermediate computations of the model by forcing similar representations for the inputs having the same execution trajectories. The proposed technique can be directly applied to the following algorithms of the CLRS benchmark: Minimum spanning tree algorithms, Sorting, Binary search, and Minimum.

As we mentioned in the previous section, any predefined trajectory can be suboptimal for a given architecture. To design our alternative approach, we make an observation that helps us to combine the advantages of learning algorithms only from input-output pairs with inductive bias improving the size generalization capabilities of the models.

---

[1]We note that for models without hints there is no abstract meaning for encoding/decoding of the intermediate states, we use such wording only for consistency with hint-based models.

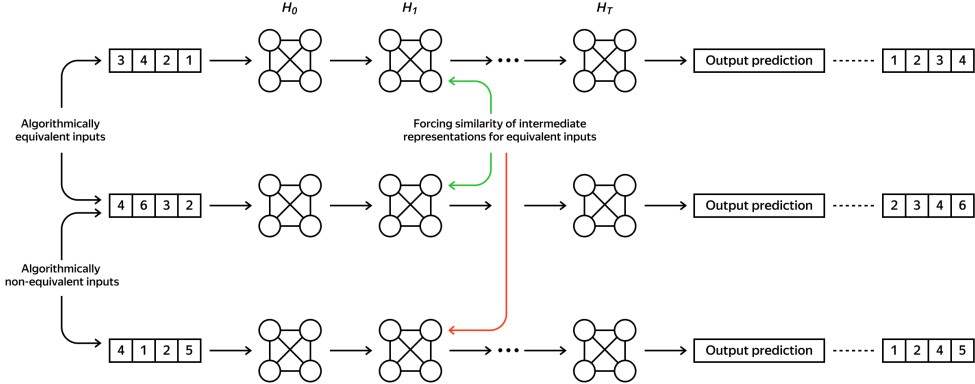

Figure 4: Self-supervised regularization for neural algorithmic reasoning.

In the absence of hints, the model does not have any knowledge about the algorithm's trajectory and its intermediate steps and potentially can align to a new way of solving the problem. Hence, obtaining any information about underlying computations of the model, for example, detecting the part of the data irrelevant to the current step, is non-trivial. Moreover, given the parallel nature of modern model architectures, all nodes can potentially be used at all steps. Thus, we propose constructing such a pair of different inputs that *every* step of a given algorithm is the same for both inputs.

Although constructing such pairs for an arbitrary algorithm is not always trivial, the underlying combinatorial structure of some tasks allows us to describe the *properties* of the desired algorithm. Such properties can reduce the search space of the model while remaining in the wide range of possible algorithms, which can solve the given problem.

For example, any comparison sort has the same trajectory for any pair of arrays with the same relative order of its elements. We clarify that we consider the sequence of the intermediate relative orders of the elements as a trajectory, not the array values themselves. A less trivial example is the minimum spanning tree problem. A wide range of algorithms (Prim (1957), Kruskal (1956), etc.) have the same execution trajectory on graphs with fixed topology (graph structure) and different weights on edges if the relative order of the corresponding edge weight is the same. As the set of all subsets of edges that do not contain a cycle forms a matroid (Harary, 1969), any greedy construction of the tree would lead to the optimal result. This observation allows us to use only the relative order of the edge weights, not the weights themselves. We note that the CLRS benchmark considers the set of edges in the tree (a mask on edges) as the result of the algorithm, not the total weight of the constructed tree.

One important remark is that for two different inputs, the exact matching of all the intermediate steps is generally a more strict condition than the matching of the result. For example, consider Kruskal's algorithm for the minimum spanning tree problem. The first stage of this algorithm is sorting the edges by their weight. The edges are then greedily added to the disjoint set of independent edges until the spanning tree is constructed. Thus, an edge with a weight that is larger than any weight from the minimum spanning tree does not affect the result but can cause the different states of the algorithm after the sorting stage.

Now we describe our idea more formally. Given a problem $P : \mathcal{X} \to \mathcal{Y}$, we define a set $A$ of the algorithms solving this problem, and a decomposition of the set of inputs $\mathcal{X} = \bigcup \mathcal{X}_k$ to the disjoint equivalent classes, such that for every two elements $X_i, X_j \in \mathcal{X}_k$ from the same class $\mathcal{X}_k$, all algorithms from $A$ produce the same execution trajectory. We note that the properties of the input data that we mentioned above produce such a decomposition of the input spaces for the specified tasks. Now we can use this to reduce the search space for our model to the space of functions acting the same way on any pair of inputs from the same equivalence class, being invariant under input changes that do not affect the algorithm's trajectory. Our idea is illustrated on Figure 4.

In contrast to Hint-ReLIC, we do not decompose each snapshot of the execution to the relevant and non-relevant nodes, as the dynamic of such decomposition is defined by the hints sequence and thus may inherit its disadvantages. Instead, we highlight the property of the whole input of the algorithm.

This idea allows us to give the model additional useful signals about the data without any explicit or implicit enforcing to any predefined dynamic, which can be suboptimal for a particular model architecture.

## 4.2 Adding contrastive term

Now, let us describe how classic contrastive learning techniques can be applied for this purpose. As mentioned above, for a given task, our goal is to learn representations of intermediate steps $t \in [1 \ldots T]$, which are similar for any pair of inputs from the same equivalence class. One natural way to do this is to build a contrastive objective aimed at solving the instance discrimination task (Wu et al., 2018, Chen et al., 2020a, Mitrovic et al., 2021).

We note that there are different ways to define an instance. For example, one may consider the whole execution of the algorithm as a single classification task, where each class represents a trajectory choice. In this case, all the intermediate states of all the nodes can be used as a single instance. However, as the algorithm's execution can be described as a step-by-step decision-making process, a more fine-grained approach is suitable, defining each node representation at each step as a separate instance. We focus on the latter option.

Let us denote by $\mathcal{X}_X$ the class of equivalence for $X \in \mathcal{X}$, which is a set of valid augmentations for $X$, under which the similarity of representations for the corresponding nodes at each step can be directly enforced. For this purpose, we use the following contrastive regularization term:

$$L_t = -\sum_{X \in \mathcal{X}} \sum_{i=1}^{n} \sum_{X_a \in \mathcal{X}_X} \log \frac{\exp(\phi(f_t(X, i), f_t(X_a, i)))}{\sum_{j=1}^{n} \exp(\phi(f_t(X, i), f_t(X_a, j)))},$$

where $f_t(X, i)$ denotes the representation of $i$-th node ($i \in [1 \ldots n]$), obtained at step $t$, and $\phi(x, y)$ denotes a comparison function for the obtained representation, which we implement as $\phi(x, y) = \langle g(x), g(y) \rangle$, where $g$ is a fully-connected neural network. We compute this term for each step $t \in [1 \ldots T]$ and add it to the output loss.

We note that this term is similar to the contrastive part of the Hint-ReLIC objective with some differences. First, in Hint-ReLIC, certain hints are contrasted (such as intermediate prediction of the predecessor in the sorted order for sort algorithms, namely $pred\_h$) to be invariant under changes that affect only those parts of the data, which are not included to the current step of execution. Such an approach allows for highlighting the importance of different nodes at different execution steps. Our method does not specify any additional details about intermediate computations, so the whole latent representations of each node state are forced to be invariant under valid augmentations. Another difference is in the augmentations choice. While Hint-ReLIC is compatible with a wide range of interventions, such as connecting the underlying graph to any randomly generated subgraph, our augmentations for $X$ are defined by the equivalence class $\mathcal{X}_X$ in which by design all the inputs have the same size. To summarize, we propose a different way of applying the standard contrastive learning technique to the algorithmic reasoning setup. While Hint-ReLIC demonstrates an advanced way to align the model to the particular trajectory of the execution, our method can be applied as additional regularization to the model, which does not favor any particular trajectory from the wide range of possible algorithms.

## 5 Experiments

In this section, we evaluate the performance of the proposed modifications to the no-hint mode and test the impact of contrastive regularisation on OOD generalization. To deepen our analysis, we also conducted additional experiments, evaluating the impact of the step count and demonstrating the influence of the contrastive term on similarity of trajectories for algorithmically equivalent inputs.

### 5.1 Experimental setup

**Baselines**   We compare our approach with several forms of hint usage. First, we consider the standard supervision of the hints, denoted as *Baseline* (Ibarz et al., 2022). Then, *No hint* is the model that completely ignores the hint sequence. Finally, *Hint-ReLIC* utilizes an advanced way of using the

Table 1: Training details and model hyperparameters

| | |
|---|---|
| Train sizes | 4, 7, 11, 13, 16 |
| Validation size | 16 |
| Test size | 64 |
| Optimiser | Adam |
| Learning rate | 0.001 |
| Train steps count | 10000 |
| Evaluate each (steps) | 50 |
| Early-stopping patience (steps) | 500 |
| Batch size | 32 |
| Processor | Triplet-GMPNN |
| Hidden state size | 128 |
| Number of message passing steps per processor step | 1 |

hints (Bevilacqua et al., 2023). All models use the Triplet-GMPNN architecture proposed by Ibarz et al. (2022).

**Model and training details** We train our model using the Adam optimizer with an initial learning rate $\eta = 0.001$, with batch size 32, making 10,000 optimization steps with early stopping based on the validation performance. Our models are trained on a single A100 GPU, requiring less than 1 hour to train. We report the results averaged across five different seeds, together with the standard error. We report all hyperparameters in Table 1.

We note that our proposed modification of the no-hint mode allows for the use of any type of representation replacing the hint prediction. For example, each sorting algorithm in the CLRS benchmark has a unique description of the intermediate computations inspired by the algorithm itself. Each step of the Bubble sort algorithm is described by the current elements order and two indexes for the current comparison. At the same time, hints for the Heapsort have a more complex structure that defines the current state of the heap and other details (Veličković et al., 2022). Such differences in the specifications of the algorithms produce differences in computational graphs for models trained for Bubble sort and Heapsort. While the proposed modification of the no-hint mode leaves freedom for such architecture choices, we consider a search of the optimal specifications (description of the representation of the input, intermediate steps, and output) for each algorithm beyond the scope of our work. Moreover, we do not use the original specifications for the hints from the CLRS benchmark, as their design (combination of types of hints) can potentially align the model to the original algorithm even without the supervision of the hints. Thus, we only construct a single type of intermediate representations to demonstrate the advantage of the proposed modification in the simplest way. We use a latent representation of type mask located at edges and use them for all the algorithms.

We also need to set the trajectory length (the number of message-passing steps) for our model. For the regime with hints, the number of steps can be obtained from the original trajectory. However, for the no-hint version, there are several options: take the number of steps from some algorithm, use a predefined steps count, or add a separate model that predicts if the algorithm execution is ended. We use the steps count linearly depending on the input size, with factors 1 and 5 for array and graph problems, respectively.

**Data augmentations** Recall that our proposed regularisation technique is aimed at forcing the invariance of the execution under the choice of the elements from the same equivalence class. To achieve this, we augment the inputs by sampling from the same data distribution, preserving the relative order of the elements for the array-based problems and the relative order of the corresponding edges for graph problems. We generate one augmentation for each data point.

**Dataset** We test our modifications of the no-hint mode on the same subset of the algorithms from the CLRS benchmark as in Bevilacqua et al. (2023). We additionally evaluate our contrastive objective on the Sorting, Minimum, Binary search, and Minimum spanning tree problems.

Table 2: Comparing performance for different hint usage modes. The table shows the mean and standard error for the test micro-F1 score across different seeds.

| Algorithm | No hint | Baseline | Hint-ReLIC | No hint (ours) | Add contr. (ours) |
|---|---|---|---|---|---|
| Artic. points | 81.97 ± 5.08 | 88.93 ± 1.92 | **98.45 ± 0.60** | 94.55 ± 2.47 | — |
| Bridges | 95.62 ± 1.03 | 93.75 ± 2.73 | **99.32 ± 0.09** | 98.91 ± 0.47 | — |
| DFS | 33.94 ± 2.57 | **39.71 ± 1.34** | — | 34.52 ± 1.17 | — |
| SCC | 57.63 ± 0.68 | 38.53 ± 0.45 | **76.79 ± 3.04** | 66.38 ± 7.12 | — |
| Topological sort | 84.29 ± 1.16 | 87.27 ± 2.67 | **96.59 ± 0.20** | 94.58 ± 0.60 | — |
| Bellman-Ford | 93.26 ± 0.04 | **96.67 ± 0.81** | 95.54 ± 1.06 | 95.25 ± 0.58 | — |
| BFS | 99.89 ± 0.03 | 99.64 ± 0.05 | 99.00 ± 0.21 | **99.95 ± 0.04** | — |
| DAG SP | 97.62 ± 0.62 | 88.12 ± 5.70 | 98.17 ± 0.26 | **98.18 ± 0.99** | — |
| Dijkstra | 95.01 ± 1.14 | 93.41 ± 1.08 | **97.74 ± 0.50** | 96.97 ± 0.32 | — |
| Floyd-Warshall | 40.80 ± 2.90 | 46.51 ± 1.30 | **72.23 ± 4.84** | 42.16 ± 2.14 | — |
| MST-Kruskal | 92.28 ± 0.82 | 91.18 ± 1.05 | **96.01 ± 0.45** | 92.89 ± 0.92 | 93.15 ± 1.01 |
| MST-Prim | 85.33 ± 1.21 | 87.64 ± 1.79 | **87.97 ± 2.94** | 85.70 ± 1.44 | 85.20 ± 1.77 |
| Insertion sort | 77.29 ± 7.42 | 75.28 ± 5.62 | 92.70 ± 1.29 | 90.67 ± 8.22 | **98.74 ± 1.12** |
| Bubble sort | 81.32 ± 6.50 | 79.87 ± 6.85 | 92.94 ± 1.23 | 90.67 ± 8.22 | **98.74 ± 1.12** |
| Quicksort | 71.60 ± 2.22 | 70.53 ± 11.59 | 93.30 ± 1.96 | 90.67 ± 8.22 | **98.74 ± 1.12** |
| Heapsort | 68.50 ± 2.81 | 32.12 ± 5.20 | 95.16 ± 1.27 | 90.67 ± 8.22 | **98.74 ± 1.12** |
| Binary Search | **93.21 ± 1.10** | 74.60 ± 3.61 | 89.68 ± 2.13 | 85.29 ± 4.52 | 87.12 ± 2.23 |
| Minimum | 99.24 ± 0.21 | 97.78 ± 0.63 | 99.37 ± 0.20 | 99.98 ± 0.02 | **99.99 ± 0.01** |

Table 3: Comparing test performance of the no-hint (ours) model on the sorting problem for different processor step counts.

| Steps count | Test score |
|---|---|
| Linear (same as for Insertion sort from CLRS) | 90.67 ± 8.22 |
| Quadratic (same as for Bubble sort from CLRS) | 90.97 ± 8.03 |
| Linearithmic (same as for Quick sort from CLRS) | 91.05 ± 8.17 |
| Linearithmic (same as for Heap sort from CLRS) | 89.66 ± 7.40 |

## 5.2 Results

Table 2 compares the performances of several hint usage modes, including those proposed in Sections 3 and 4.[2] We first note that our modification of the no-hint mode significantly improves the performance compared to the standard *No hint* version, achieving the best-known results on the BFS, DAG shortest paths, and Minimum problems. For instance, for sorting, the performance increases from 68.5%–81.3% for *No hint* to 90.7% for our version. Here we note that such a spread in performance between different sorting algorithms in the standard no-hint mode is caused by different step counts: *No hint* does not rely on hints, but uses a particular algorithm when setting the number of message-passing steps. In contrast, our modification uses a pre-defined step count depending on the input size. Also, compared to baseline supervision on hints, our no-hint model yields better performance on 14 out of 18 tasks. To conclude, our simple modifications of the no-hint mode already demonstrate competitive performance on most of the problems.

The last column of Table 2 demonstrates further improvements obtained due to the proposed self-supervised regularization. In particular, with this regularization, we achieve the best results for sorting (98.7%), significantly outperforming the existing models.

We also evaluate the effect of the step count for the sorting problem, see Table 3. For our experiments, we used the same step count as the Baseline model for Insertion sort. We demonstrate that mimicking the step counts from other baseline models still shows the advantages of the proposed no-hint changes.

---

[2]The Hint-ReLIC results for DFS are not reported since Bevilacqua et al. (2023) achieve the perfect score with a simpler technique.

Finally, we show that for the sorting problem, the proposed contrastive term forces the model to have similar representations for inputs from the same equivalence class, as intended. For this purpose, we took different pairs of equivalent inputs and for each step of the execution we calculated the following ratio: if we take one node from one input and take the closest node from the augmented input, what is the probability that the closest node will represent the element with the same relative order. Figure 5 shows that while the model without the contrastive term (green line) is able to mainly rely on the relative order, our contrastive term is forcing much more confidence (and speed) in achieving that property.

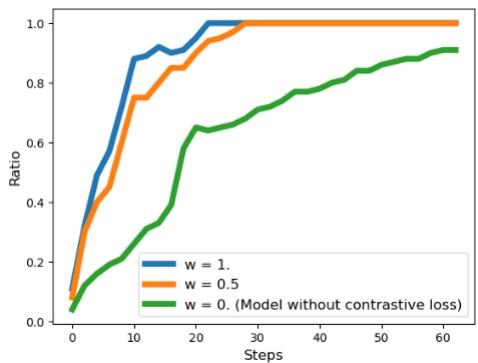

Figure 5: Influence of the contrastive term on similarity of the representations of equivalent inputs depending on weight factor $w$ of the contrastive term. Insertion sort problem, test data (64 nodes).

## 6 Limitations

In this paper, we propose improvements of the no-hint regime for neural algorithmic reasoning and a method to regularise intermediate computations without enforcing the model to any predefined algorithm trajectories. While this approach seems natural and promising, some limitations remain.

Our contrastive learning approach applies to the limited set of problems. While for many algorithms it is possible to construct the inputs that produce the same trajectory, sometimes it can be non-trivial. For example, for the shortest path problems, linear transformation of each weight does not affect the resulting paths and all intermediate computations (such as the order of inserting nodes to the heap for the Dijkstra's algorithm (Dijkstra, 1959)). However, such augmentations are not informative due to the layer normalization, and constructing more diverse examples represents a separate task.

While the proposed self-supervised regularization aims to reduce the search space for the model, it does not give a model a strong inductive bias towards underlying computations. We believe that more advanced techniques can be useful for algorithms with long sequential reasoning, such as the minimum spanning tree. Developing stronger inductive biases is a promising direction for future research.

For our self-supervised regularization, we augment the inputs by sampling from the same data distribution. While this gives a signal to the model that it should account only for the relative order, this signal can be even more significant if we use different distributions. Investigating the effect of diverse data distributions on the performance of neural algorithmic reasoners is a good subject of a separate study.

## 7 Conclusion

In this work, we propose several improvements to learning algorithms without intermediate supervision. We also demonstrate the possibility of regularising all computational steps of the model without enforcing it to any predefined algorithm trajectory. The obtained models outperform previous no-hint models and show competitive performance with models that have access to the entire algorithm trajectory. We also achieve new state-of-the-art results on several tasks. In particular, a considerable improvement is obtained for the sorting algorithms, where we get the score of $98.4\%$.

We consider neural algorithmic reasoning without intermediate supervision a promising direction for further advances in neural algorithmic reasoners.

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
