# OpenReview forum: "Neural Algorithmic Reasoning Without Intermediate Supervision"
_NeurIPS.cc/2023/Conference — NeurIPS 2023 poster_

### Official Review · Reviewer_y7mK · 2023-06-29

**Soundness:** 3 good
**Presentation:** 2 fair
**Contribution:** 3 good
**Rating:** 6
**Confidence:** 2

**Summary:**

The paper addresses neural algorithmic reasoning without the supervision of intermediate steps of reasoning. Typically, neural algorithmic reasoning requires supervision on the intermediate steps of reasoning. The paper proposes a method that does not require intermediate supervision and achieves competitive results with full-supervised benchmarks. The paper proposed a modified no-hint reasoning scheme that mimics the hints-supervision setting and self-supervised objective using the contrastive loss term for nodes for each step.

**Strengths:**

Overall the paper addresses an important problem of the integration of reasoning with neural networks.
The motivation is clearly argued in the paper. Neural algorithmic reasoning has been showing its strong capacity for structured reasoning as neural networks, but the approach requires dense supervision on intermediate steps. The proposed method mitigates this issue, reducing the amount of required effort and thus would extend the capacity and applicability of neural algorithmic reasoning for different tasks.
The proposed method achieves competitive results against fully supervised benchmarks and shows its advantages. The obtained results are interesting and promising. The source code is available in the supplementary material.

**Weaknesses:**

My major concerns regarding this paper are as follows.
- The explanation of the method is somewhat unclear and is not easy to follow. For example, Hint-ReLIC is repeatedly referred to and used as an explanation. However,  the paper does not provide enough explanations regarding Hint-ReLIC (although textual explanations are distributed in different parts of the paper), and that makes the presentation not easy to follow.  Readers would gain benefits from a bit more detailed explanations regarding the key relevant method in the background section to understand the proposed method.
- Experimental details are not explained enough.  In Table 1, some cells do not contain values, but no specific reasons are clarified in the main text (if I am not missing them). In the right-most column of Table 1 and Table 2, values for Insertion/Bubble/Quick/Heap sort are all the same, even though the experiments are conducted on 5 different random seeds and stochastic gradient descent. This seems to imply that the proposed method produces exactly the same values for each different random seed across different tasks via stochastic optimization. If there is an explanation for why this happens, it would need to be mentioned in the main text.

**Questions:**

Why are some values not available in Table 1?

In the right-most column of Table 1 and Table 2, values for Insertion/Bubble/Quick/Heap sort are all the same, even though the experiments are conducted on 5 different random seeds and stochastic gradient descent. Why does this happen?

**Limitations:**

Some limitations are argued in the paper.

---

> ### Author Rebuttal · Authors · 2023-08-09
>
>
> Thank you for your thoughtful review!
>
> We agree that adding a more detailed explanation of the Hint-ReLIC method to the background section would make the paper easier to follow, we will update this section. However, we are happy and open to clarifying something during the discussion period if needed.
>
> We address your questions below.
>
> > Why are some values not available in Table 1?
>
> For the Hint-ReLIC method, the DFS scores were not reported by Bevilacqua et al. (2023) (as we mention in the footnote on page 7). For the last column, as mentioned in line 180 of the paper, our proposed regularization term is applicable only for part of the problems, which is also mentioned as one of the main limitations in Section 6.
>
> > In the right-most column of Table 1 and Table 2, values for Insertion/Bubble/Quick/Heap sort are all the same, even though the experiments are conducted on 5 different random seeds and stochastic gradient descent. Why does this happen?
>
> As the proposed no-hint regime does not use any information about the algorithm, all sorting tasks become the same and scores may differ only by a random seed, so we reported the same numbers for all sorting algorithms. But, as we mention in lines 303-306 of the paper, the standard no-hint regime uses the ground-truth step count for each algorithm, which produces the difference for sorting algorithms in the first column.
>
> We hope that our response addresses your concerns and will be happy to answer any additional questions during the discussion period if needed.

---

> > ### Comment · Reviewer_y7mK · 2023-08-14
> > **Reply to the rebuttal**
> >
> > I thank the authors for the detailed clarification.
> >
> > As the concerns are addressed adequately, I raise the score to 6 (from 5).
> > I defer to the other reviewers for the deep assessment of the novelty/originality of this work due to my limited knowledge.

---

### Official Review · Reviewer_Fw9t · 2023-07-05

**Soundness:** 2 fair
**Presentation:** 2 fair
**Contribution:** 1 poor
**Rating:** 4
**Confidence:** 4

**Summary:**

This paper addresses the challenge of developing neural networks capable of performing algorithmic reasoning. The paper discusses disadvantages about the use of intermediate hints during training of algorithmic reasoners. The authors propose a regularisation term to ensure that meaningful representations are learnt during the neural execution of algorithms, even when no supervision occurs on intermediate steps.

**Strengths:**

The paper is written clearly. Discussion of background and prior work is comprehensive. Source code is provided for reproducibility of the experiments. The paper is not very original, as it borrows concepts and formulations from [1]. However, the authors use these concepts in an alternative way.

While [1] uses the regularisation loss to build a causal model, the authors use it to ensure that a neural network computes the same representations (at all steps) for any two inputs x_1, x_2 for which the execution of the algorithm A would be the same. I find this idea a clever augmentation technique. Quantitative results show supposedly good improvements in sorting algorithms with respect to baselines.

[1] B. Bevilacqua, K. Nikiforou, B. Ibarz, I. Bica, M. Paganini, C. Blundell, J. Mitrovic, and P. Velickovic. Neural algorithmic reasoning with causal regularisation.

**Weaknesses:**

As mentioned above, all concepts and regularisation terms are not new and borrowed as-is from [1]. Experimental details are somewhat scarce and sometimes imprecise. For instance, the authors do not state the hidden dimensions of their models. Furthermore, this technique seems to be applicable only on a small portion of all problems in the CLRS benchmark, which greatly limits the potentiality of the method. Fundamentally, this technique seems to be only useful in the case of sorting, since in all other tested techniques it performs significantly worse. Plus, training networks the way authors propose in this paper will not align with any specific algorithm dynamics (as the authors correctly recognised in the limitation). In fact, judging from the sorting results (98.74% everywhere) it seems that the network has learnt a different algorithm w.r.t to insertion/bubble/quick/heap sort (or possibly only one of them).

The authors main point is that the regularisation term is forcing similar representations for the inputs to have the same execution trajectories, but there’s no experiments nor theoretical analysis confirming this property.

One thing I find very concerning is that the authors claim to fix the number of steps of the network (i.e., the number of steps in the “neural execution” phase) to be linearly dependent on the number of input size (i.e., O(n)). This means that the network must have learnt a sorting algorithm that achieves around 98% accuracy and runs in linear time, which feels very strange, unless the constant is high enough to behave well with array of 64 elements (test size).

[1] B. Bevilacqua, K. Nikiforou, B. Ibarz, I. Bica, M. Paganini, C. Blundell, J. Mitrovic, and P. Velickovic. Neural algorithmic reasoning with causal regularisation.

**Questions:**

Q1) Can the Authors provide some insights on why the proposed technique only works for sorting algorithms?

Q2) The proposed technique seems likely to be effectively used in conjunction with supervision on hints: have the Authors tried that? Or do the Authors believe that it is not applicable?

Q3) Given that the manuscript reports 98.74% \pm 1.12 for all sorting algorithms, it looks like that the network is not aligning with any specific sorting algorithm, but potentially learns a different one. Have the Authors tried investigating in this direction?

Q4) What is the exact number of steps that the network performs during algorithm rollout?

**Limitations:**

Limitations are adequately addressed.

---

> ### Author Rebuttal · Authors · 2023-08-09
>
> We thank the reviewer for a thoughtful review of our paper!
>
> First, we would like to emphasize one of our contributions - the modification of the no-hint mode towards being more similar to the hint-based one - that is not discussed in the review. While this contribution is technically simple, we believe it has great importance, as a simple modification, which is applicable for all problems, demonstrates clear (14 of 18) disadvantages of direct hint supervision ("Baseline" vs "No-hint (ours)" columns in Table 1). This contributes to the main takeaway of our paper that no-hint training is competitive to the hint-based one and thus requires more attention from the community.
>
> We address the questions and concerns below.
>
> > all concepts and regularization terms are not new and borrowed as-is from [1]
>
> We note that the concept and the regularization from [1] is a standard contrastive technique, see, e.g., [2], [3]. However, the main contribution of [1] is the application of this technique to algorithmic reasoning in a particular way. The way this technique is applied in [1] has a significant difference from our work: [1] uses the contrastive term to align intermediate computations of the model to the ground-truth dynamics of the algorithm, while our contrastive term gives the model additional signal about the problem without providing any constraints about computations which the model needs to follow.
>
> > experimental details are somewhat scarce and sometimes imprecise. For instance, the authors do not state the hidden dimensions of their models
>
> While we used the same setup as in the past works, we agree that all experimental details should be clarified in the text. We listed our hyperparameters in Table 1 in the [global response](https://openreview.net/forum?id=vBwSACOB3x&noteId=UUd9q6foXC) (please, see the attached pdf).
>
> > training networks the way authors propose in this paper will not align with any specific algorithm dynamics (as the authors correctly recognised in the limitation)
>
> A model without hints indeed may not align with any algorithm dynamics, but we consider this as a desirable property, since
> - Aligning to a particular algorithm can be suboptimal (lines 136-139 of the paper)
> - Allowing the model to find optimal dynamic can lead to new ways to solve the problem (lines 140-142)
>
> > There’s no experiments nor theoretical analysis confirming that the regularisation term is forcing similar representations.
>
> To address this comment, we conducted an additional illustrative experiment. Please, see our global response and Fig. 1(b) in the attached pdf.
>
> > One thing I find very concerning is that the authors claim to fix the number of steps of the network (i.e., the number of steps in the “neural execution” phase) to be linearly dependent on the number of input size (i.e., O(n)). This means that the network must have learnt a sorting algorithm that achieves around 98% accuracy and runs in linear time
>
> We note that each processor's step is the message-passing, which operates (in the case of Triplet-GMPNN) over the full graph, so in terms of the problem size the computation complexity is $O(n^3)$ (which is the upper bound for all considered algorithms), so there is no need to use high constant, we used 5 for graph problems, and 1 for others. For more clarity, we used the exact same steps count as the original hint trajectory for Insertion sort, so this choice seems reasonable.
>
> Other options for step count are described in lines 286-288 of the paper.
>
> We also include ablation on the step count for sorting (Table 2 from the global response).
>
> Q1) In our opinion (which reflects the second limitation from Section 6) the reason is that invariance under order-preserving shifts reduces space differently for different tasks. For example, learning the sorting algorithm from information about the relative order is almost trivial, while solving the MST problem having the relative order over edge weights still requires additional logic, such as checking cycles between edges, adding edges to the tree, etc., which is not forced by the proposed regularization. We consider developing additional/stronger inductive biases as a promising direction for future work.
>
> Q2) We have not tried that and it seems applicable, but it does not reflect the key points of our paper: 1. The usual comparison hint vs no-hint is inaccurate due to the significant computation graph differences; 2. For some problems, it is possible to give a model additional inductive bias without any predefined constraints about underlying computations (such as the original algorithm trajectory).
>
> Nevertheless, we believe that these two directions (hints vs no-hints) can converge to the point with a balance between strong inductive biases and the absence of intense supervision, which is closely related to your idea.
>
> Q3) We would like to refer to Fig. 1(a) from the global response, which demonstrates that due to the parallel architecture, the model without hints is making most of the final predictions in a parallel way, which differs from the sequential nature of the sorting algorithms.
>
> Q4) As mentioned above, it is $5 n$ for the graph problems and $n$ for the array problems.
>
> We are happy to discuss further any of the raised weaknesses and questions!
>
> [1] B. Bevilacqua, K. Nikiforou, B. Ibarz, I. Bica, M. Paganini, C. Blundell, J. Mitrovic, and P. Velickovic. Neural algorithmic reasoning with causal regularisation.
>
> [2] Mitrovic, J., McWilliams, B., Walker, J. C., Buesing, L. H., and Blundell, C. Representation learning via invariant causal mechanisms
>
> [3] Ting Chen, Simon Kornblith, Mohammad Norouzi, and Geoffrey Hinton. A simple framework for contrastive learning of visual representations

---

> > ### Comment · Reviewer_Fw9t · 2023-08-17
> > **Considerations on rebuttal**
> >
> > First of all, thanks to the Authors for their response and for clarifying previously obscure experimental details.
> >
> > Let there be no doubts, both contributions (i.e., modification of the no-hint regime and addition of the self supervised term) are sensible and interesting investigations, especially considering targeting of NP-hard problems, where we clearly cannot supervise on intermediate hints.
> >
> > However, there are still fundamental issues in the work:
> > 1) From the authors’ response it has emerged that data for all algorithms are cliques (even for sorting algorithms were usually data are arrays and could very well be represented as undirected chains), this should be specified clearly in the text. This is a direct consequences of using the Triplet-GMPNN processor. As being fully-connected, cliques suit the ``parallel processing’’ paradigm that the authors present in lines 136-139 as a motivation for introducing the self-supervised term. That is also the reason why their model could perform sorting in O(n) steps. The performance of the method was not tested in case where algorithmic data are represented differently from cliques, or, equivalently, if a different processor was used (not based on fully-connected graphs), making the assessment of the proposed approach not possible in such cases. Surely, one can always take into account the fully-connected version of any graphs when applying this method, but this would {\it greatly} impact scalability. The method would be inapplicable for large graphs.
> >
> > 2) Figure 1(b) shows that the self-supervised term does bring representations closer, for nodes having the same algorithm trajectory. However, I think that the authors must include also other baselines in that analysis, such as Hint-ReliC, and move the figure somewhere in the main paper. It is important to compare the behaviour of node representations also for other baselines, especially considering that GNNs in general tend to bring neighboring nodes' representations closer anyway. I expect the authors' proposed method still performs the best, but I believe it has to be verified and illustrated clearly.
> >
> > 3) Since the model is not aligning with any specific algorithms, it is not fair to present results on Heapsort, Quicksort, Bubblesort and Insertion Sort as four different results. Indeed, without supervising on hints these four algorithms are indistinguishable from one another. This is confirmed by the fact that the model achieved exactly the same accuracy on all four algorithms, as I pointed out in the original Q3. Overall, this method seems to tackle {it solving of algorithmic problems} rather than {\it learning of algorithms}. One may argue that in order to {\it solve} algorithmic problems, the network must learn an algorithm anyway. While I agree with this, I still find the comparisons within Table 1 not sensible, since those results show how well algorithms are imitated in principle.
> >
> > 4) Follow-up question from 3): I suspect that authors' used the same 5 seeds for all experiments on sorting, since results are * exactly *​ the same and algorithms are indistinguishable given the absence of supervision on intermediate steps. For the same reason, I believe the same should happen for MST algorithms (Kruskal and Prim), but results are different there. Did the authors use different seeds for Kruskal and Prim?

---

> > > ### Author Response · Authors · 2023-08-18
> > >
> > > Thank you for your involvement in the discussion and valuable comments!
> > >
> > > Let us address the remaining concerns:
> > >
> > > Q1) Let us explain why we believe that our contributions do not rely on the full graph.
> > >
> > > First, parallel processing arises even with the mentioned chain graph inputs, as a single message-passing step can simultaneously update all nodes at once. We also note that even on the chain graph, sorting could perform in O(n) steps on the message-passing framework with global graph features (the presence of the global state requires only one additional node with $n$ additional edges). For example, in the insertion sort, the insertion phase for each node could be done in $O(1)$ (as proposed by the original hints trajectory from the CLRS Algorithmic Reasoning Benchmark, where hints were designed for parallelizable but not necessarily fully-connected architectures, such as, e.g., a chain).
> > >
> > > Second, the proposed contrastive term is also not relying on the density of the graph, as this term can be used with any architecture.
> > >
> > > Since both our ideas (no-hint modifications and contrastive term) can be applied to any processor,  we have chosen Triplet-GMPNN as the state-of-the-art processor showing good results in the previous work (Ibarz et al., 2022; Bevilacqua et al., 2023). We agree that the details of Triplet-GMPNN should be added to the experiment setup and will do this.
> > >
> > > Q2) We are grateful to the reviewers for their ideas for additional experiments and will definitely add all the figures and tables from the general response to the updated version of the paper. We would be happy to use the Hint-ReLIC models for additional experiments, however, the original implementation of this method is not published yet. We will add other hint-based sorting models (Bubble sort, Merge sort, Quicksort) as baselines to this comparison.
> > >
> > > Q3) It is true that without hints different algorithms become indistinguishable from one another and it is more suitable to consider the original problem (e.g., sorting), rather than a particular way of solving it. But we note that the alignment of the model to the particular execution trajectory is usually not the goal, but mainly the approach to improve the generalization abilities of the models (Veličković et al., 2022). So the comparison of the OOD performance for different methods (such as, for example, aligning to Bubble Sort with hint supervision, to Merge Sort with Hint-ReLIC, or training the sorting problem without hints) is reasonable and crucial on the path to strongly generalizable reasoners. We also agree that demonstrating no-hint performance under the names of the particular algorithms can be confusing, however, such choice is used as a convention in previous work (Mahdavi et al. (2023), Bevilacqua et al., (2023)). We will clarify this in the text by adding the corresponding comment for the table. If you have any other suggestions for how to better present this result - we will be happy to incorporate them in the revised version.
> > >
> > > Q4) Yes, as all the sorting tasks become the same and scores may differ only by a random seed, so we reported the same numbers for all sorting algorithms to highlight this. However, for the MST problem, while the underlying problem is the same and the same input/output can be used, there are differences coming from the CLRS Algorithmic Reasoning Benchmark: the Kruskal algorithm takes the graph as an input and the output is a binary mask on the edges from the MST, while the Prim algorithm takes as input the graph and the starting node and predicts for each node the pointer to another node (representing the order of extension of connected components). Such differences produce separate optimization problems.
> > >
> > > We would like to know whether we have answered all the questions and concerns raised by the reviewer. We are happy to discuss further any of the raised weaknesses and questions!

---

> > > > ### Comment · Reviewer_Fw9t · 2023-08-20
> > > > **Further response**
> > > >
> > > > A1) I acknowledge that sorting can be performed, theoretically, in O(n) processor steps even in the case of a chain graph. I expect, however, a degradation in performance compared to the full graph version (whether this degradation would be significant is to be verified through experimentation). Parallelism is indeed preserved, but the network would be obviously unable to perform as many comparisons as it does for the full graph version.
> > > >
> > > > A2) I am happy to hear about adding more baselines to the mentioned experiments.
> > > >
> > > > A3) In my previous question, I was not only referring to the no-hint version but also to the model with the contrastive term. I acknowledge that the authors followed prior work in presenting results of Table 1. However, I strongly encourage the authors to clarify this in the text.
> > > >
> > > > A4) I see.
> > > >
> > > > I thank the authors for clarifying many of my concerns and for engaging in a very fruitful discussion. I consider, however, the lack of experiments on non-fully connected graphs a major drawback of the empirical evaluation of the proposed approach. In light of this, I decided to raise my score but I am not fully convinced that this work, without a stronger empirical evaluation, deserves a full accept.

---

> > > > > ### Author Response · Authors · 2023-08-21
> > > > >
> > > > > We thank the reviewer for their deep involvement in the discussion and for raising the score!
> > > > >
> > > > > Regarding a major drawback, mentioned by the reviewer:
> > > > >
> > > > > We would note that currently there is a significant gap in performance between full-graph and sparse architectures, making the dense architectures the objects of the main interest in the field: Mahdavi et al. (2023) - MPNN over the full graph combined with transformer over the full graph; Bevilacqua et al., (2023) - Triplet-GMPNN; Ibarz et al. (2022) - Triplet-GMPNN; Diao and Loynd, (2022) - transformer over the full graph. For example, Ibarz et al. (2022) demonstrates >= 25% performance gain over other architectures on average across different algorithms.
> > > > >
> > > > > Even given the fact that our contributions do not rely on the full graph and improve the current state-of-the-art performance, we agree that proposed changes can affect the performance differently depending on the architecture. We would be happy to consider sparse architectures in our evaluation and will be happy if you mention some architectures you would like to see in such experiments. To the best of our knowledge, there are no sparse architectures with performance close to the current state-of-the-art models.

---

### Official Review · Reviewer_uiKS · 2023-07-06

**Soundness:** 2 fair
**Presentation:** 2 fair
**Contribution:** 3 good
**Rating:** 5
**Confidence:** 3

**Summary:**

The paper is about neural algorithmic reasoning, which is the task of building models that can execute classical algorithms. The paper focuses on learning algorithms without intermediate supervision, which means using only input-output pairs and not the steps (hints) of the algorithm. The paper proposes two improvements for this setting: a simple architectural change that makes the model more similar to the hint-based version, and a self-supervised objective that regularizes the model by forcing similar representations for inputs with the same execution trajectories. The paper shows that these improvements lead to competitive or state-of-the-art results on a subset of the CLRS benchmark. The paper argues that learning algorithms without intermediate supervision is a promising direction for neural algorithmic reasoning.

**Strengths:**

1. Learning algorithms without intermediate supervision is an important direction for neural algorithmic reasoning.
2. The contrastive term added is insightful and the corresponding results are promising.

**Weaknesses:**

1. The fundamental difference between the proposed "no-hint" and the original "no-hint" has not been stated clearly - the proposed "no hint" seems to only have added the hidden layer size or the neural network capacity. If so, the comparison against the original “no-hint” is unfair. Can the authors give a more detailed description of the proposed “no hint” architecture? Also, please list relevant hyperparameters (like size, layers) on the network, and conduct a comparative experiment on the network size.
2. The “step count” setting in section 5.2 is also confusing: is step count the main reason why the proposed no-hint performs better than the original no-hint?
3. Table 2 is somewhat confusing: No hint (with our updates) & Binary Search has 93.21 ± 1.10 which is different from 87.12 ± 2.23 or 85.29 ± 4.52  in  Table 1. Also, I think taking the maximum of No hints (ours) and Add contr. (ours) as “ours” in Table 2 is not a good choice since they are different methods proposed.
4. Although the proposed new contrast item can improve the performance without hints, I doubt that the design of this item is extremely challenging (which might have already been mentioned in section 6 limitations), even more so than obtaining hint data. This is because the design of this item requires a deep understanding of the target algorithm itself and the designer must be an algorithm expert, while obtaining hint data only requires a corresponding executable algorithm. This may deviate from the original intention of “without intermediate supervision” and greatly reduce the actual effectiveness of this paper.
5. The writing needs to be improved.

**Questions:**

1. Table 2 should be merged into Table 1 since only one column is different.
2. Can you show the latent representation change with the number of steps? Can it correspond to the hint of the algorithm?

**Limitations:**

Limitations have been discussed by the authors.

---

> ### Author Rebuttal · Authors · 2023-08-09
>
> We thank the reviewer for a thoughtful review! Let us address the raised concerns and questions.
>
> > Can the authors give a more detailed description of the proposed “no hint” architecture?
>
> The main difference between the original no-hint version and the proposed one is the presence of an encoding-decoding stage after each processor step. The difference is demonstrated in Figure 2 of the paper and motivated by the presence of such stages in hint-based models. Such modifications add a small group of parameters for the encoder (green line in Figure 2) and decoder (red line in Figure 2) (encoder and decoder are single linear layers), keeping the hidden size of the model the same.
>
> > The comparison against the original “no-hint” is unfair
>
> One of the key points of our paper is that the original (used in previous work) comparison hint vs no-hint, which demonstrated the advantages of using hints, is unfair due to the significant computation graph differences (left and middle parts of Figure 2). Our proposed modification makes the comparison hint vs no-hint more accurate (middle and right parts of Figure 2) and demonstrates the disadvantage of direct hint supervision over the no-hint version for most of the problems (columns "Baseline" and "No-hint (ours)" in Table 1).
>
> > Also, please list relevant hyperparameters (like size, layers) on the network, and conduct a comparative experiment on the network size.
>
> Model sizes and other hyperparameters are nested from the previous work. To address your question, we specified all the hyperparameters in Table 1 of the [global response](https://openreview.net/forum?id=vBwSACOB3x&noteId=UUd9q6foXC). Also, since our contribution is not a development of new model architecture, we believe that experiments with the network size are orthogonal to our work. However, we added some other ablations to the global response.
>
> > The “step count” setting in section 5.2 is also confusing: is step count the main reason why the proposed no-hint performs better than the original no-hint?
>
> No, for example, for sorting, we used the same step count as in the original Insertion sort hint trajectory ($n$ steps for input with $n$ nodes). Also, to answer this question, we conducted an ablation on the steps count for sorting (Table 2 from the global response), demonstrating almost the same performance for different step counts. The motivation behind our choice was to avoid an input-dependent number of steps (as, for example, for Heap sort) as it is unnecessary for the no-hint mode. So we made the simplest choice which is applicable to all the problems at the same time.
>
> > the design of this item requires a deep understanding of the target algorithm itself and the designer must be an algorithm expert, while obtaining hint data only requires a corresponding executable algorithm
>
> We definitely agree that finding augmentations is not trivial and requires a deep understanding of the algorithm and will describe it in the limitations section. However, we also note that hints don’t come “for free” with dataset generator, at least, the useful ones: Veličković et al. (2022) noted some difficulties with hint generation, such as compression (simple ‘for’ loops can be described as a single step for parallel architectures) or ability to predict the next hint from the current one, which can represent a separate task for different architectures. Also, the same hints can be encoded/predicted differently, which can affect the model performance. While we believe that for most of the tasks and architectures useful hint sequences can be found, the comparison between simple supervision on hints and no-hint mode (“Baseline” and “No hints (ours)” columns in Table 1 of the paper) shows that for most of the problems (14 of 18) no-hints performance is better (or better hints are needed). We also note that invariants/augmentations from the paper need to be discovered/developed once for a task, while helpful hints may need to be engineered separately for different architectures (line 131 in the paper).
>
> > The writing needs to be improved.
>
> We would be grateful if you could clarify which aspects of writing need to be improved, then we will revise the paper accordingly.
>
> > Table 2 should be merged into Table 1 since only one column is different.
>
> Our motivation for including a separate table was to use the aggregated numbers depending on the different hint usage modes, for highlighting the progress of no-hint and hint-based reasoners. While such a comparison of different methods can be imprecise, it serves as a proxy to answering the question: Given fixed model size and different strategies of hint usage, which strategy is preferable for different tasks? We think that combining two tables could be more confusing for readers as some results from Table 2 were taken not from Table 1, for example, for the DFS problem (see the footnote on page 7 of the paper).
>
> > Table 2 is somewhat confusing: No hint (with our updates) & Binary Search has 93.21 ± 1.10 which is different from 87.12 ± 2.23 or 85.29 ± 4.52 in Table 1.
>
> As we mentioned above, the motivation behind Table 2 was to aggregate the best results depending on hint usage mode. Binary search is the only problem that did not gain performance with our updates, so the best performance without hints is still 93.21 ± 1.10, which is taken from the first column.
>
> > Can you show the latent representation change with the number of steps? Can it correspond to the hint of the algorithm?
>
> Thank you for the suggestion! We refer to Fig. 1(a) in the global response.
>
> We would like to know whether we have answered all the questions and concerns raised by the reviewer. We are happy to discuss further any of the raised weaknesses and questions!

---

> > ### Comment · Reviewer_uiKS · 2023-08-17
> >
> > Thanks for the response! Some of my concerns have been addressed but I still have the following ones:
> >
> > **1. About the "no-hint" structure.**
> >
> > The authors say that "Such modifications add a small group of parameters for the encoder (green line in Figure 2) and decoder (red line in Figure 2) (encoder and decoder are single linear layers), keeping the hidden size of the model the same." So my original understanding is correct, right? "no hint" seems to only add two linear layers, and the effect is to increase the network capacity.
> >
> > Or, can the authors explain more about the design of the "encoder and decoder"? There is no evidence why the added layers are named "encoder and decoder" because there is no demarcation between the so-called encoder and decoder like corresponding loss, explicit tensor form transformation, etc. Think about seq2seq models, the demarcation between the encoder and decoder is very clear. As a comparison, I cannot see the demarcation in "no hint". And that's why I take the "encoder-decoder" design as simply increasing network capacity and ask the authors to list hyperparameters of the network.
> >
> > Or, one simple question: What does the latent representation look like and how do you supervise the training of it? "Hint" is not necessary here but you must have some constraints like losses or architecture designs or training algorithms, to make the latent representation behaves like a hint. Otherwise, they are simple linear layers.
> >
> > **2. The design of this item requires a deep understanding of the target algorithm itself and the designer must be an algorithm expert.**
> >
> > Can the authors give a concrete example of the design cost of the two methods? This is important for validating the motivation.

---

> > > ### Author Response · Authors · 2023-08-18
> > >
> > > Thank you for being involved in the discussion and for all your questions, which highlight the parts of the paper that need to be additionally clarified!
> > >
> > > **1. About the "no-hint" structure.**
> > >
> > > >So my original understanding is correct, right? "no hint" seems to only add two linear layers, and the effect is to increase the network capacity.
> > >
> > > Yes, the difference is just in two linear layers, which are presented in the model with hint supervision. In other words, we argue that for a fair comparison versus the hint model, one needs to remove the loss on the intermediate steps, keeping the computational graph the same, while the previous work removed these linear layers (Figure 2 of the paper).
> > >
> > > >Or, can the authors explain more about the design of the "encoder and decoder"?
> > >
> > > Such naming is nested from the model with hints, the encoder maps abstract objects (node features, edge weights) to the latent space, while the decoder constructs abstract objects (e.g., a pointer from a node to another node) from the latent space. We agree that for a no-hint model, this naming is useful only for encoding inputs and decoding outputs of the model, while for the intermediate steps there is no abstract meaning. Also, phrases like «we propose to run the no-hint model in the encoded-decoded mode» (line 161) from the paper are used only for consistency with hint-based models. We will make it more transparent in the text.
> > >
> > > >What does the latent representation look like and how do you supervise the training of it?
> > >
> > > Such a «decoded» intermediate state is a single logit for each edge (line 283) (binary mask over the edges), which is «encoded» to the adjacent node features at the next step of the computation. Such states have no additional constraints:
> > >
> > > - the no-hint model is trained only with supervision loss on the output of the model (and not the intermediate steps);
> > >
> > > - the proposed contrastive term forces invariance of these states for equivalent inputs, but these states are not supposed to ‘behave like a hint’ (while potentially could).
> > >
> > >
> > > **2. The design of this item requires a deep understanding of the target algorithm itself and the designer must be an algorithm expert.**
> > >
> > > >Can the authors give a concrete example of the design cost of the two methods?
> > >
> > > First, we would like to note that we do not argue that finding the invariances is in some sense easier or harder than building good hints, both methods have additional costs. But the main difference between these two methods is not the cost, but that the invariance utilizes the fact that models usually find solutions that are more aligned with their architectures (so aligning them to a particular trajectory can be a suboptimal or difficult task).
> > >
> > > Answering your question:
> > >
> > > We believe that for hint supervision, the main cost is to align the model to the desired algorithm: for each architecture, one needs to decompose the algorithm execution trajectory to the sequence of steps, each of which the model should predict. Also, one needs to carefully design transitions between steps, making hints predictable from previous steps, but not so simple, keeping the execution trajectory relatively short. For example, we would refer to the hint trajectories from the CLRS Algorithmic Reasoning Benchmark (Veličković et al., 2022) for the Insertion Sort algorithm. The insertion phase is “compressed” to the single step of message-passing (which is possible due to the parallel processing of modern architectures), another option was to use another constant steps count or $O(n)$ steps (similar to the ground-truth algorithm), even the simple algorithm produces a lot of different ways to convert it to the hints trajectory and each particular choice directly affects the performance of the model, as intermediate predictions are supervised to correspond exactly to the hints. Given the fact that for 14 of 18 problems the no-hint (ours) performance is better than hint supervision, we could say that existing hints/hint losses are possibly not optimal.
> > >
> > > As for the proposed contrastive technique, the main cost is to understand for which inputs the algorithm will perform identically. For example, for comparison sort algorithms this is easy (we can consider inputs with the same relative order of the elements), for other algorithms it may require a deeper understanding of the underlying problem.
> > >
> > > The common idea behind the contrastive term is utilizing the same execution trajectory for different inputs. Even a simple notion on the relative order of the inputs is applicable for several different problems since sorting or taking the maximum occurs as a subtask in a variety of problems (e.g., in addition to the MST problem described in the paper, there is also building the heap/search tree from some values). Once such invariants are found, it is clear how to apply the contrastive term regardless of the architecture used.
> > >
> > > We hope this answers the question and we are open to further discussion on the raised points.

---

> > > > ### Comment · Reviewer_uiKS · 2023-08-21
> > > >
> > > > Thanks for the authors' response! I have doubts about the two questions, and I have raised my score. However, I'm still not convinced by the "we do not argue that finding the invariances is in some sense easier or harder than building good hints" and take it as a weakness of the paper.

---

### Official Review · Reviewer_P3vy · 2023-07-07

**Soundness:** 3 good
**Presentation:** 3 good
**Contribution:** 3 good
**Rating:** 7
**Confidence:** 3

**Summary:**

This paper proposes a novel method for Algorithmic Reasoning without intermediate supervision. The core idea is to use a self-supervised objective that regularize the internal computations. The authors evaluated the proposed method with CLRS algorithmic reasoning benchmark and achieve state-of-the-art performance on a subset of problems. To the best of my knowledge, the contribution is novel and clear and I recommend acceptance.

**Strengths:**

This paper is well-motivated and relatively easy to follow. The design of regularization for the self supervision scheme is also quite intuitive. The evaluations are comprehensive and seems to support the effectiveness of the proposed approach. I appreciate the specific emphasis of potential limitations.

**Weaknesses:**

While the main experiment/evaluation is informative, I would appreciate if the authors address some potential ablations such as varying numbers of augmentations.

**Questions:**

In the limitation sections, the authors mentioned the issues with requirements for strong inductive bias for the self-supervised regularization scheme. I would appreciate the authors to address the potential concern on computation with respect to search.

**Limitations:**

As mentioned in Strength, I appreciate authors' detailed inclusion of limitations. I do not think there are major limitations in addition to the said ones given the current content.

---

> ### Author Rebuttal · Authors · 2023-08-09
>
> We thank the reviewer for the review and positive feedback!
>
> Following your suggestion, we conducted several ablations (please, see the details in the [global response](https://openreview.net/forum?id=vBwSACOB3x&noteId=UUd9q6foXC)):
>
> - Different augmentations count (Table 3 from the global response). Keeping the batch size equal to 32, we tried different distributions of positive and negative examples for each element in the batch, and used them in the contrastive term, as described in lines 242-248 of the paper. This experiment shows that the method is not sensitive to a particular choice of the augmentations count. For the results in the paper, we used only one positive example, and the other 30 were used as negative examples.
> - Different processor steps count for sorting (Table 2 from the global response).
> - Experimental verification of stronger invariance of the model trained with the contrastive term (Fig. 1(b)).
>
> Answering your question, we also would like to clarify our thoughts on stronger inductive biases for other problems. The core reason why the proposed contrastive term improves the performance differently for different tasks is that the invariance under order-preserving shifts reduces the space differently for different tasks. For example, learning the sorting algorithm from information about the relative order is almost trivial, while solving the MST problem having the relative order over edge weights still requires additional logic, such as checking cycles between edges, adding edges to the tree, etc., which is not forced by the proposed regularization. That is why we consider developing additional/stronger inductive biases as a promising direction for future work. Does that answer your question? We are happy and open to further discussion on this topic.

---

### Official Review · Reviewer_Sqsb · 2023-07-09

**Soundness:** 2 fair
**Presentation:** 4 excellent
**Contribution:** 3 good
**Rating:** 7
**Confidence:** 4

**Summary:**

The paper focuses on neural networks that learn to execute algorithms, using the CLRS benchmark. It tackles the case of learning to execute when there are no hints available (no access to intermediate steps of the algorithm's trace). To do that, it proposes two modifications: the first is architectural, which is maintaining the latent representation of the current step and using it for the next step, which previous no-hint architectures discard and architectures using the hints use to decode the predicted hint and compute an additional loss by comparing with the ground-truth hint. The second one proposes a contrastive loss aimed to reduce the function search space, by using the following invariance: inputs that have the same (algorithmic) trajectory (belonging to the same equivalence class should have the same representation. The two modifications improve on the no hints baseline and in many cases match the model with hints.

**Strengths:**

As the strengths outweigh the weaknesses of this paper, I recommend acceptance in its current form. I would be happy to increase my score if some of the weaknesses are addressed.

The strengths of the paper include great clarity and sensible proposals to improve no-hint performance, such as maintaining a similar computation graph and using invariances to reduce the search space in lack of stronger inductive biases. Moreover, the evaluation is thorough and the improvements brought show a clear step in the direction of accurately learning algorithms without hints.

**Weaknesses:**

I think the paper would be greatly strengthened by improving the motivation towards doing no-hint learning. As the performance of the proposed models is not consistently better than that using hints, it is likely that at the moment, this is not enough evidence to convince towards porting to no-hint neural algorithmic reasoners for polynomial algorithms.

One way to improve this would be to improve it empirically (considering other invariances, introducing stronger inductive biases, extending the number of algorithms it is applicable to—perhaps applying the model to NP-hard problems, where having hints is not scalable because of exponential-time complexity), or providing ablation studies of some of the initial motivating points: abilities of no-hint architectures to do parallel processing or new ways/insights of learning to solve the problem.  I also think that the argument of human effort should be further clarified in the limitations — while hints are generated using the algorithm's implementation therefore they come "for free" with the dataset generator, finding augmentations requires understanding of the algorithm's invariances, which is also based on human effort.

It would be useful if a few phrases could be clarified:
- “We note that the computation of hint predictions and encoding them to the processor in the encoded-decoded mode can be expressed as an additional message-passing step over the nodes, edges, and graph features” could be made clearer by including information on how hints are computed
- "a latent representation of type mask located at edges” is not immediately obvious
- it is mentioned that the setup is the same as in previous work, but restating the training and testing regime (such as size of inputs) would be beneficial.

**Questions:**

Please see weaknesses section.

**Limitations:**

The authors have included a discussion on the limitations of the proposed method.

---

> ### Author Rebuttal · Authors · 2023-08-09
>
> We thank the reviewer for their review and constructive comments!
>
> We would like to additionally support our motivation towards investigating no-hint reasoners with an experiment (Fig. 1(a) from the global response), which demonstrates the execution trajectories for sorting models. We discuss our observations in the [global response](https://openreview.net/forum?id=vBwSACOB3x&noteId=UUd9q6foXC). In particular, we observe that models with hints may struggle to learn the relation between the intermediate predictions of the pointers and the output predictions. Also, the model without hints has a much stronger tendency to do parallel processing. Such insights combined with common intuition about the redundancy of over-engineering (as models usually find solutions that are more aligned with their architectures) represent the core of our motivation.
>
> Also thank you for your notes on human effort for hints/augmentations - we definitely agree that finding augmentations is not trivial and requires a deep understanding of the algorithm and we will describe it in the limitations section. However, we also note that hints don’t come “for free” with dataset generator, at least, the useful ones: Veličković et al. (2022) noted some difficulties with hint generation, such as compression (simple ‘for’ loops can be described as a single step for parallel architectures) or ability to predict the next hint from the current one, which can represent a separate task for different architectures. Also, the same hints can be encoded/predicted differently, which can affect the model performance. While we believe that for most of the tasks and architectures useful hint sequences can be found, the comparison between simple supervision on hints and no-hint mode (“Baseline” and “No hints (ours)” columns in Table 1 of the paper) shows that for most of the problems (14 of 18) no-hints performance is better (or better hints are needed). We also note that invariants/augmentations from the paper need to be discovered/developed once for a task, while helpful hints may need to be engineered separately for different architectures (line 131 in the paper).
>
> We also thank the reviewer for noticing parts that could be clarified in the text, we will describe them in more detail in our updated version. However, we are happy and open to clarifying something during the discussion period if needed.

---

> > ### Comment · Reviewer_Sqsb · 2023-08-19
> >
> > Thank you for the response! The main point I raised has been partially addressed through one of the new studies, strengthening my opinion that the paper should be accepted. I increased my score accordingly.

---

### Author Rebuttal · Authors · 2023-08-09

We thank all the reviewers for their feedback and many constructive suggestions and comments.

We have incorporated additional clarifications and experiments in the attached pdf, which contains:

**Table 1.** Parameters of the model and training procedure.

**Table 2.** Ablation on the steps count for the sorting problem. In the paper, we used the same steps count as the Baseline model for Insertion sort. We demonstrate that mimicking the steps count from other baseline models still shows the advantages of the proposed no-hint changes.

**Table 3.** Ablation on the augmentations count for the contrastive term.

**Figure 1(a).** Demonstration of the execution dynamics for the hint-based model vs the no-hint model on the sorting task.

While the interpretation of the underlying computations of a model is non-trivial, we can use the decoders of models to demonstrate the dynamics of the intermediate predictions. We took two models for the sorting problem: No-hint (ours) and Insertion sort (trained with hints supervision). Both models use the same steps count (64 steps for the test data). After each processor step, we can apply the output decoder to node embeddings to see the intermediate predictions of the predecessor (we remind that the output for the sorting problem is represented by the prediction of the predecessor node in the sorted order). For each step, we visualize the ratio of the intermediate pointers that are equal to the final output of the model.

We note that for hint-based models we have two decoders: the Hint decoder and the output decoder, both of which can be used for this purpose (orange and blue line, respectively).

We would like to highlight several insights from this experiment:

1. Models trained with supervision on the trajectory for Insertion sort struggle to learn the relation between the intermediate predictions of the pointers and the output predictions (orange line). In other words, the model potentially can consider the hint sequence (and even each transition between hints) as a separate task, not related to the output of the algorithm directly. However, the execution trajectory for the models with hints inherits some sequential order of intermediate updates (blue line).

2. The model without hints has a much stronger tendency to do parallel processing - almost all output predictions are obtained on the first steps of the execution (green line).

**Figure 2(b).** We demonstrate that the proposed contrastive term forces the model to have similar representations for inputs from the same equivalence class. For this purpose, we took different pairs of equivalent inputs and for each step of the execution we calculated the following ratio: if we take one node from one input and take the closest node from the augmented input, what is the probability that the closest node will represent the element with the same relative order (and not, for example, the closest value)?

While the model without the contrastive term (blue line) is able to mainly rely on the relative order, we see that our contrastive term is forcing much more confidence (and speed) in achieving that property.

We are happy to discuss these experiments further or answer any questions!

---

### Decision · Program_Chairs · 2023-09-21

**Decision:**

Accept (poster)

**Comment:**

Neural algorithmic reasoning aims at training neural networks that are capable of imitating classical algorithms. These methods often use the intermediate state of the algorithm as supervision to improve the representation learning and the final performance. This of course, create a limitation and makes it difficult to use these methods out-of-the-box for a new algorithm. This paper proposes a method to remove intermediate hint supervision. The method however does rely using augmentations and a contrastive loss which should be carefully designed and depend on the specific target algorithm. That being said, demonstrating this alternative seems very useful. Overall most reviewers find this work interesting and voted for acceptance. However, there are several existing concerns as well and I suggest authors take those concerns seriously and improve the camera-ready version of the paper. Improving the quality would certainly encourage more people to build on this work.

A few areas to improve for camera-ready:

- Specifying all experiment details and choices in the appendix for all experiments. Everything should be reproducible from reading the paper.
- Adding experiments on at least one other graph family (see reviewer Fw9t's comments)
- Add ablations to see if some of the improvement of the proposed architecture can be attributed to extra capacity added to the network. Authors can then discuss what would be a "fair" comparison but having those results allow the reader to make up their own mind after seeing all the experimental results and reading authors' arguments.
-  For some of the CLRS algorithm discuss how easy it is to come up with invariances as opposed to using hints so that the reader can understand this issue better (see reviewer uiKS's comments)

This is a borderline paper. I am voting for acceptance and hoping that authors would understand that taking the above comments seriously would ultimately benefit their own paper and the community.